# Proposal for a new meteotsunami intensity index.

Clare Lewis[1][2], Tim Smyth[2], Jess Neumann[1], Hannah Cloke[1][3]

[1] Department of Geography & Environmental Science, University of Reading, Reading, UK

[2] Plymouth Marine Laboratory, Prospect Place, Plymouth, Devon, PL1 3DH, UK

[3] Department of Meteorology, University of Reading, Reading, UK

*Correspondence to*: Clare Lewis (clare.lewis@pgr.reading.ac.uk)

**Abstract**

Atmospherically generated coastal waves labelled as meteotsunami are known to cause destruction, injury and fatality due to their rapid onset and unexpected nature. Unlike other coastal hazards such as tsunami, there exists no standardised means of quantifying this phenomenon which is crucial for understanding shoreline impacts and to enable researchers to establish a shared language and framework for meteotsunami analysis and comparison.

In this study, we present a new 5-level Lewis Meteotsunami Intensity Index (LMTI) trialled in the United Kingdom (UK) but designed for global applicability. A comprehensive dataset of meteotsunami events recorded in the UK was utilised and the index's effectiveness was evaluated, with intensity level and spatial distribution of meteotsunami occurrence derived. Results revealed a predominant occurrence of Level 2 moderate intensity meteotsunamis (69%) in the UK, with distinct hotspots identified in Southwest England and Scotland. Further trial implementation of the LMTI in a global capacity revealed its potential adaptability to other meteotsunami prone regions facilitating the comparison of events and promoting standardisation of assessment methodologies.

**1 Introduction**

If you live in a coastal zone, you are at risk from being impacted by various hydrometeorological hazards, one such hazard is the meteorological tsunami or meteotsunami. This is a globally occurring shallow water wave which tends to be initiated by sudden air pressure changes and wind stress from moving atmospheric systems such as convective clouds, cyclones, squalls, thunderstorms, gravity waves and strong mid-tropospheric winds (Vilibic´ and Šepic, 2017). The atmospheric disturbance transfers energy into the ocean initiating and amplifying a water wave that then travels towards the coastline where it is further amplified through coastal resonances (Šepic et al. 2012). There are a range of geneses associated with meteotsunami, however air pressure change has been the traditional dominant factor in the generation and propagation of this phenomenon worldwide. Certain meteotsunami can be driven by strong wind fronts, as exemplified by the 'winter type meteotsunamis' in the Northern Baltic Sea (Pellikka et al, 2022). Infra gravity waves linked to strong mid tropospheric jets are also correlated with

meteotsunami genesis, this is restricted to such locations as the Mediterranean, Chile and Australia but not so prevalent in the Tropics (Zemunik, 2022). According to Denamiel et al (2023) infra gravity waves manifest as rapid surface pressure oscillations and low sea level pressure. This reinforces Pellikka et al (2022) and Rabinovich (2020) who state that air pressure plays a dominant role in some of the world's strongest meteotsunami. Following on from this and with the data available we examined the source of each event and chose air pressure as the primary atmospheric component to cover both mid latitudes and equatorial regions to, allowing for global standardisation of the index.

Due to the rapid onset and unexpected nature of these waves, they have the potential to pose a considerable threat to coastal communities, infrastructure and ecosystems (Sibley et al. 2016). This has been apparent throughout recent history with an increase in the number of meteotsunami being experienced around the world. With extreme events such as those in Vela Luka (Croatia, 1978) where a 6m wave caused US$7 million damage; at Nagasaki (Japan, 1979) where an event killed three people; Dayton Beach (Florida, 1992) where a single 3 m wave injured 75 people and caused damaged to dozens of cars and the Persian Gulf (2017) where a squall line initiated a 2.5 m wave leaving 22 injured and five dead (Gusiakov, 2021).

Understanding the intensity and impact of meteotsunami is crucial for effective coastal hazard management. The development of the LMTI index involved an extensive review of existing global meteotsunami scales and indices to which it was found that there is an absence of a working methodology. There was an initial suggestion at an intensity scale for meteotsunami as presented in an editorial by Vilibic´ et al (2021). However, as acknowledged by the author this scale was limited to the events and papers presented in the special edition and was designed to represent a feature that might be used for cataloguing meteotsunami. There is no detailed methodology available for this index, the scoring appears to be based upon wave height, injuries and fatalities. While fatalities can indicate the severity of an event, they are influenced by a range of factors. Using fatality as a sole aspect would mean that a meteotsunami arriving on the shores of a highly populated area would indeed have more of an impact than an event occurring in a less populated area. It assumes that an event is only of high intensity if it has an anthropogenic impact.

Due to the absence of a working intensity index, we subsequently reviewed tsunami scales and indices, as these have a similarity to meteotsunami in wave types and impacts. The review revealed two types of indices used for defining and quantifying tsunami:

- A magnitude scale which relates to the physical quantities and parameters of the hazard including the source of the event and/or the wave height (Imamura-Iida scale, 1967). These scales tend to be logarithmic, and this allows for the compression of a wide range of values into a smaller range. This makes it easier to compare and visualise data that spans several orders of magnitude. However, it can make it difficult to translate the results to a non-academic community. Magnitude scales tend to compare only the wave size and not it's strength.

- An intensity scale that assesses the impacts of an event, including expected damage, based on observations (Papadopoulos and Imamura scale, 2001). It is easier to interpret and compare than other scales and can incorporate the human element without instrumentation. However, its reliance on descriptive evidence can lead to subjective results.

In this paper, we present a novel approach to assessing meteotsunami intensity by introducing a new 5 level meteotsunami
intensity index named the Lewis Meteotsunami Intensity Index (LMTI). We provide an overview of the development process
and implementation of this index, focussing on its application in the UK as a case study with a view to further global
applicability.
**2 Index development**
Creating the LMTI involved four stages, (Figure 1).
**2.1 Stage 1: Catalogue of events**
Trials for the LMTI were conducted in the UK, where there is a long history of events dating back to at least 1750 AD (Haslett
and Bryant 2009). Six main sources of UK meteotsunami events were utilised: Lewis et al. (2023), Williams et al. (2021),
Thompson et al. (2020), Long (2015), Haslett and Bryant (2009) and Dawson et al (2000) all providing a comprehensive and
coherent historical record. The collected data were analysed, with the meteotsunami identified and categorised according to a
reliability and verification system adopted from Gusiakov (2021). Identified events were allocated a reliability score from 1 to
4 depending on the amount of evidence and data available across the sources (i.e., the number of components completed in the
index), where 1= doubtful (1 to 3 components), 2= questionable (eyewitness report, 3 to 6 components), 3= probable
(newspaper report, 6 to 9 components) and 4= definite (technical report, 9 to 12 components). Older events which are usually
fragmented make it difficult to establish an informed judgement, so these were subsequently allocated a reliability score of 1;
events with insufficient information remained unclassified and were considered highly uncertain.
**2.2 Stage 2: Meteotsunami components and values**
The proposed LMTI considers 12 various components of meteotsunami and receptor site characteristics, based upon
descriptions of previous global events, current thresholds used by researchers and the characteristics of other related hazard
indices (Table 1). This multifaceted approach allows for the LMTI to capture the complex dynamics of meteotsunami events
and facilitate a single score which can be matched with a description on the LMTI index table (Table 1). The LMTI adopts
this layout to allow for intensity evaluation based upon hazard only or receptor site only. By incorporating both parts this
allows for analysis of a low height wave impacting a highly vulnerable coastline. Each component has a different threshold
weighting leading to the allocation of a score from 1 to 5. These threshold weightings are calculated based on event data and
other related hazard indices.

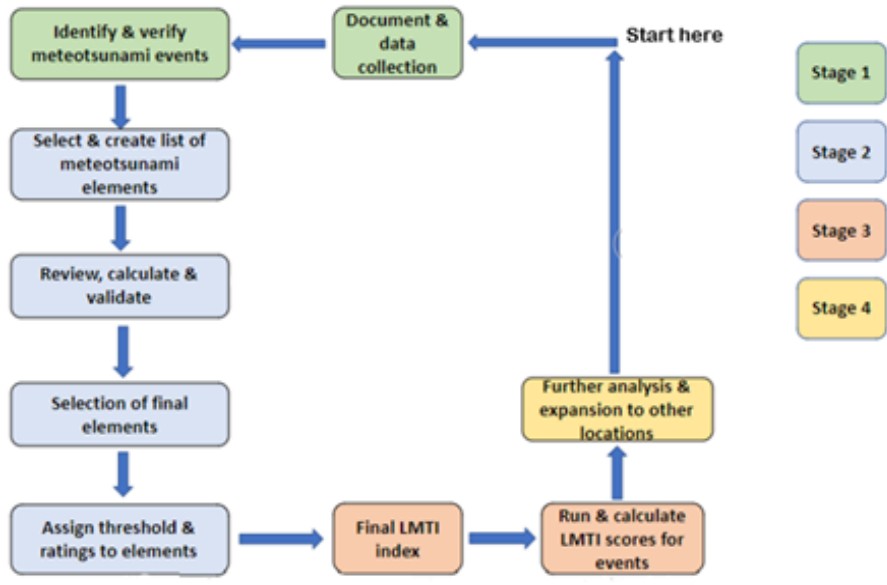

Figure 1: The adopted methodology for the development of the LMTI.

**2.2.1 Physical hazard characteristics**

**Maximum wave height (Mw):** the vertical distance between wave trough and crest (m) at the shoreline. This is the most frequently used element when discussing tsunami and meteotsunami (Williams et al. 2021, Gusiakov 2021) as wave height is the easiest form of data to observe. The greater the wave height, the greater the volume of water impacting people and structures along the shoreline. A wave height threshold of 0.30 m or less was selected as the baseline for Level 1 (minimal intensity), which was decided by analysing average wave heights of global and UK events, where 0.3 m was found to be the threshold for potential damage (Lynett et al. 2014).

**Currents (Cr):** the velocity (m/s) of the water's movement produced by the meteotsunami wave as it inundates the shoreline. The faster the current the more the displacement of people, animals, and debris. The values for LMTI are based upon those laid out in Lynett et al. (2014) for tsunami waves which is calculated upon not only past event data from buoys and boats but also from experienced eyewitness accounts and videos.

**Maximum inland intrusion of seawater (Fd):** the inland extent (m) of seawater flow past the high tide mark. The further inland the water reaches, the higher the risk to assets. However, this can be restricted by local topography which is addressed in subsection 2. This component can contribute to the impact of an event through flooding and as such is frequently used in coastal flooding indices (Rocha, Antunes and Catita, 2020). Including this component in the index allows for comparisons of events and provides a comprehensive and quantifiable measure of the potential damage and impact. This provides information for decision makers to assess the role of local topography in the extent of flooding impact.

**Additional or compound hazards (Ch):** considerations of other hazards linked to the source system and their potential to
elevate the overall level of risk are considered in this component, one point is accumulated for each additional hazard that
occurs parallel to the meteotsunami event. Existing tsunami indices do not include this component as it is deemed an external
factor. However, we feel that due to the interactive nature of meteotsunami with other hazards, it is imperative that it be
considered. The risk from meteotsunami is not just restricted to elevated water level and velocity, if coupled with hazards such
storm surge, seiching, precipitation water levels may become elevated. High winds, mudflows, and lightning can produce
compound issues.  This element also covers wind dynamics such as intense gusts, windstorms and abrupt changes in direction
which can also generate significant wave energy.
**Air pressure change (Ap):** the rate of change in the localised air pressure (mb) within a 3-minute period. This is included as
a key component in the initiation of a meteotsunami via the inverse barometer effect and has been found to be present in many
of the world's strongest meteotsunami. The sharper the air pressure changes the greater the potential for water displacement, 1
mb change equals 1cm change in static water level. The thresholds for this component have been derived from the data recorded
from global events which range from 0.5 to 1.5 mb in approx. 3 minutes. This air pressure change creates a connection with
the mechanisms in the open sea, which can significantly affect the amplification or attenuation of meteotsunami waves. By
using air pressure as a component researchers can identify commonalities and differences between regions in relation to one
of the principle forcing mechanisms.
**Tidal regime (Ti):** the tidal stage at the time and location of maximum wave impact at the shoreline. This can be either neap,
spring, low, mid or high. Coastal areas experiencing a spring or high tide are characterised as being highly vulnerable with the
impacts being exacerbated by an already elevated water level. Whilst the authors acknowledge the importance of tidal range
in coastal dynamics within this category, after a provisional analysis of tidal ranges in locations prone to different intensity
meteotsunami, we could not find a direct correlation. However, it was found that coastal infrastructure in certain regions such
as the Mediterranean, is adapted to the local tidal range and as such the effects of meteotsunami are modulated by this. In
micro tidal areas whilst the wave energy is lower it tends to be more concentrated and in macro tidal areas where the wave
energy is a lot stronger the impact tends to be dissipated over a larger area.

**2.2.2 Receptor site characteristics**

**Time of arrival of maximum wave at the shoreline (Pw):** the time of day at the location of maximum wave activity and is
sub divided into approximately 3-hour slots. This element is imperative to assessing the risk to human life. The highest scoring
category (5 = extreme) equates to the most likely time of day where people, assets and commercial activity will be present
along the shoreline.
**Shoreline geomorphology (Sm):** the composition of the dominant shoreline material type. The five classes are scored
accordingly based on the erosion capability of water, relative resistance, and the ability of the material to diffuse wave power
and alter the flow characteristics. The five classes of shoreline material range from the fastest and least resistant material of a
sandy beach (5 points); bedrock and gravel shores (4 points); estuarine and vegetated zones (3 points); artificial frontage such
as concrete seawalls (2 points) and finally to hard igneous rocks (1 point), scoring for this component is adapted from Masselink
et al. 2020 and Gornitz, 1991. For this component, geomorphic classes were defined based on a visual interpretation of the
shoreline material in the immediate area of inundation using high resolution satellite imagery (Google Earth).
**Shoreline gradient (Sg):** the steepness of the coastal zone (°) and is linked to the susceptibility of the area to inundation and
flooding by meteotsunami waves. This component can inform decision makers on mitigation factors that may need to be
implemented. The thresholds created for this index are adopted from the vulnerability index of Gornitz (1991) which is an
already accepted and implemented methodology for assessing coastal hazards and risk. The gentler the slope the greater the
loss of land to seawater and the higher the vulnerability. This is defined as the ratio of altitude change to the horizontal distance
between any two points in the coastal hinterland behind the initial elevation and is calculated using Google Earth as a distance
finder and then by applying the following calculation Eq. (1):

$$Sg = \frac{\text{Hsl}}{\text{Pd}} \, 100 \qquad\qquad (1)$$


Where Hsl represents height above sea level in (m) of the selected feature point. Pd is the straight point distance from 0 m
above sea level to a point of interest such as a hospital, school, or park.
**Shoreline elevation (Se):** average height (m) above sea level of the area in the immediate vicinity of the shoreline. The
thresholds are again based on the vulnerability index of Gornitz (1991) where the elevation zone within 5 m of the shoreline
faces the highest probability of inundation. The higher the elevation values the less vulnerable the area to inundation, as
elevation provides more resistance to water flow. This can be calculated by using an online elevation finder (freemaptools.com)
and is the average of six random elevation points within a 1000 m zone of the mean high-water spring (MHWS) level enabling
measurement during all tidal stages. In considering the possibility of redundancy amongst 'Sg' and 'Se' components, a cursory
assessment was carried out using the well-established variance inflation factor or VIF. A VIF value of 1.162 was yielded, this
indicates that the two components are not correlated. If the VIF value was higher and nearer to '5' this would suggest that the
index may have redundant components or overlapping information.
**Asset impact (Ai):** This is one of two qualitative components present in the index, and it represents the level of flooding and
disruption experienced on infrastructure, historical, ecological, agricultural, livestock and property at the location. With scoring
ranging from no impacts to minor (short term inconvenience and disruption), moderate (repairable), to severe (structural
damage with interruption of critical infrastructure) to extreme (long term damage where assets are lost and written off).
**Fatality and/or injury (Fi):** This is the second qualitative component and accounts for the number of individual fatalities and
general injury to persons in the affected area as a direct result of the event. If we measured meteotsunami intensity solely in
terms of loss of life this would be an inaccurate approach as it does not consider the hazard but rather just one aspect of its
impact. This component and the one preceding it were included to assess the level of asset damage and to allow for long term
trend analysis. If fatality levels at a certain location start to drop after the implementation of a warning system, this will indicate
that the system has been effective. With this component 'minor' relates to only cuts and bruises experienced, 'moderate' relates
to broken bones and non-permanent trauma, 'severe' is permanent damage to a limb or organ and 'extreme' is fatality.

**2.2.3 LMTI intensity levels**

Once the thresholds were determined it was possible to then propose a five-stage index. This system incorporates a scoring
regime to represent the level of contribution or weighting from each component towards the overall hazard. For this reason,
each component is scored separately on a level of 1 to 5, with 1 contributing least and 5 contributing most strongly. This
method allows for standardisation of the index and for each component that is measured in different units to be combined.
Papadopoulos and Imamura (2001) proposed a 12-level scale to measure tsunamis, however, we have reduced and simplified
the LMTI scale to 5 levels, as meteotsunamis, being smaller in scale and more localised in impact than tsunamis, do not need
such a detailed breakdown.
The final meteotsunami intensity values exhibited in Table 2 contain brief descriptions highlighting the characteristics of each
intensity level which have been devised from the characteristics of historical global meteotsunami events and are based around
the events ability to be measured, its impacts and post event actions. The five levels are portrayed in a colour coded format as
this is an effective way of communication as people tend to perceive risk better through colours, graphics, and visuals (Engeset
et al. 2022).

**2.3 Stage 3: Categorising events based on intensity: How to calculate LMTI**

1.   An event must be identified and verified as a meteotsunami (see Lewis et al. 2023).
2.   The 12 components are systematically allocated a score of 1 to 5 dependant on the distinct weightings of the

207         threshold values as displayed in Table 1.

The component scores from each of the two subsections are added together and divided by the number of
component cells containing data.  If a component is not present at certain locations, then the numerical score of
'zero' is placed in the calculation and this does not affect the overall intensity score.
3.   Scores for the two subsections are then combined to give a single score by using the following conceptual

212         calculation Eq. (2):

$$MTI = \frac{\Sigma z}{Nz} \tag{2}$$


Where LMTI (meteotsunami intensity) is a function of 12 potential components, where Z is component and N

216       is the number of components.

4.  The final LMTI score will be a number between 1 and 5 as shown in Table 2 and will give a standardised

218        description of the level of intensity for that event. The higher the intensity score the higher the level of risk.


In the calculation of the index the scores are expressed with decimal places as shown in supplementary 1 for example, LMTI
1.3 or LMTI 3.4, this enables a fine resolution for quantifying and comparing intensity and impact for research purposes. The
presentation of the final intensity score is represented as a whole number, where the index is typically rounded to the nearest
integer for example, LMTI 1 or LMTI 3. This simplified representation provides a clear categorisation to present to the public,
stakeholders and decision makers.

226                Table 1: Hazard and receptor components with associated thresholds as used in the LMTI.

| Score | | 1 (minimal) | 2 (moderate) | 3 (high) | 4 (severe) | 5 (extreme) |
|---|---|---|---|---|---|---|
| Hazard | Wm: Max wave height (m) | <0.3 | 0.3 to 0.7 | 0.8 to 2 | 2.1 to 3.9 | 4+ |
| | Cr: Currents created (m/s) | <0.75 | 0.75 to 1.5 | 1.6 to 2 | 2.1 to 4 | 4+ |
| | Fd: Max inland flooding (m) | <2 | 2 to 10 | 11 to 50 | 51 to 100 | 100+ |
| | Ch: Number of cumulative hazards | none | one | two | three | four + |
| | Ap: Air pressure change (mb/3 mins) | <0.5 | 0.5 to 0.7 | 0.8 to 1 | 1.1 to 1.9 | 2+ |
| | Ti: Tidal stage at peak wave | Neap | Low | Mid | High | Spring |
| | | | | | | |
| Receptor | Pw: Time of peak wave arrival (24 hr) | 0.00 to 05.00 | 21.00 to 00.00 | 05.00 to 10.00 | 15.00 to 21.00 | 10.00 to 15.00 |
| | Sm: Shoreline geomorphology | Rocky (Igneous) | Artifical Frontage | Estuarine (saltmarsh) | Rocky (sedimentary or metamorphic) | Sandy (beach, dunes) |
| | Sg: Shoreline gradient (%) | >20 | 20 to 10 | 9 to 5 | 4 to 1 | <1 |
| | Se: Shoreline elevation ASL (m) | >30 | 30 to 10 | 9 to 5 | 4 to 2 | <2 |
| | Ai: Asset impact (Human & Eco) | none | Flooding, minor disruption | Moderate damage | Severe damage | Large scale, Long term |
| | Fi: Fatality/Injury | none | minor injury | Moderate injury | Severe injury | Fatality |




230                Table 2: LMTI intensity level descriptions.

| MTI | Description |
|---|---|
| L1 (green) | **Minimal.** Only detectable on instruments, weak with no direct threat to life & assets, no action required. |
| L2 (yellow) | **Moderate.** Visible in instruments & observations, slight disruption, accompanied by other hazards, small debris & shallow flow, rarely a threat to life & assets. |
| L3 (orange) | **High.** Large debris, violent movement of vessels & cars parked in flood zones, multi hazard situation with frequent threat to life & assets, fast water velocity with deep water extending past flood risk defences. Future coastal plan required. |
| L4 (red) | **Severe.** Violent movement & damage to infrastructure and assets. Pollution by contaminants. Significant threat to life & assets, coastline retreat & erosion with a multi-hazard situation. Large debris in fast flowing, deep water. Significant & active adaption methods required for the future. |
| L5 (purple) | **Extreme.** Widespread & extensive threat to life & assets. Heavily damaging with long term changes to the coastal profile and ecological assets. Heavy objects washed away or moved to a higher elevation with fast and deep water. Multi hazard situation requiring extensive pre-event preparedness measures. |



**3 Stage 4: Application of the Index**
We demonstrate the practical application of the LMTI in this paper by applying the index to the combined lists of UK
meteotsunami events (Lewis et al. (2023), Williams et al. (2021), Thompson et al. (2020), Long (2015), Haslett and Bryant
(2009) and Dawson et al (2000)). The full dataset of UK results can be found in S1: supplementary information and on an
interactive map available at  https://www.google.com/maps/d/edit?mid=1RiSeW-DIPSylIVOLv_8-
T8Gy_e0To08&usp=sharing.
To further demonstrate the LMTIs practicality and to lay the groundwork for its global application, a selection of 30
worldwide events as sourced from Vilibic´ et al. (2021) and Pattiaratchi and Wijeratne (2015) had the index applied to them to
extrapolate intensity scores (S2: supplementary information). The LMTI in this format offers a valuable tool for researchers,
enabling comparative analyses between different regions and to facilitate a better understanding of meteotsunami dynamics
in a global capacity.

## 3.1 UK meteotsunami intensity

The trial run of the LMTI provided valuable insights into UK meteotsunami events. A total of 100 events were analysed,
amongst these events, Level 2 meteotsunamis accounted for 69 % of the occurrences (Figure 2). This finding suggests that the
UK is prone to moderate intensity meteotsunami. Level 1 (minimal) meteotsunamis represented 12 % of events, in particular
between 2009 and 2015. Level 3 (high) meteotsunamis accounted for 16 % of the events especially between 1883 and 1932.
Finally, the results revealed a small number of severe intensity events (Level 4) which appeared in the hazard subsection, with
all three events occurring in the winter months and along the Bristol Channel.
The results highlighted in Supplementary 1 show that the number of unreliable meteotsunamis (those classified as 1= doubtful
and 2= questionable) decreases over time, with none recorded after 1968. 67 % of the events were classified as definite
meteotsunamis having been attributed a high reliability score of 4. This enhanced reliability is apparent in the record since
2008, which is an indication of the abundance of data with increasing instrumentation.
The distribution of meteotsunami hotspots was also identified through the application. The southwest region of England
exhibited a concentration of all levels of intensity type events, with the Bristol Channel exhibiting the only Level 4 type events.
The south of England and north of Scotland also demonstrated notable meteotsunami activity in particular Level 2 (moderate)
intensity events (Figure 3). These hotspots highlight the region's most at risk from meteotsunami occurrence and provide a
valuable insight for future coastal management.

## 3.2 Global expansion of the Index

The findings from the trial implementation of the LMTI in a global context demonstrated that the index has the potential for
adoption into other coastal regions prone to meteotsunami. Results for events such as Vela Luka (Croatia) in 1978, Nagasaki
(Japan) in 1979, Ciutadella (Menorca) in 2006 all scored an expected with a Level 3 on the scale in line with the observed
data. The event in 2017 in Dayyer (Persian Gulf) scored a Level 4, this was particularly deadly as it occurred in an area that
was not accustomed to experiencing extreme wave events and so consequently the infrastructure and population were not
prepared. It occurred at 08.00 local time, a few hours after a thunderstorm and it was calm, so people were starting their day
unaware of any issue. On the LMTI index a Level 3 and 4 equates to high intensity, where large debris is deposited from high

velocity water flow and there is a threat to life and assets (Table 2). On the opposite end of the intensity scale at Level 1, corresponding to minimal intensity events which are only detectable on instruments and with no impact to life or assets, we find events such as Pellinki (Finland) in 2010. If we compare these results to the intensity scale proposed by Vilibic´ et al (2021), we find a correlation. According to Vilibic´ et al (2021) Vela Luka, Croatia (1978), Ciutadella, Menorca (2006) and Pantano do Sul, Brazil (2009) were all allocated an intensity Level 4, LMTI scored these events at a 3.8, 3.6 and 3.5 respectively. Cassino beach (Brazil) in 2014, Zandvoort (Netherlands) in 2017 and Mali Losinj (Croatia) in 2007 were allocated a Level 3 according to Vilibic´ et al, LMTI scored these events at a 3.2, 3.1 and 3 respectively. Arraial do Cabo (Brazil) in 2002 and Lagos (Portugal) in 2010) were both allocated a 2 by Vilibic´ et al, LMTI scored these events at a 2.2 and a 2 respectively. Finally, the Persian Gulf event of 2017 was allocated a Level 5 and LMTI scored it at a 4.3.

Validation was a critical step in assessing the accuracy and applicability of the index. This procedure involved ensuring that the index accurately reflected the observed and recorded data for the event that it was quantifying. For example, the well documented and researched event at Vela Luka (Croatia) on the 21 June 1978, where in the early evening the bay experienced a 6 m water level change with accompanying strong currents which inundated 650 m inland. The impact was large scale damage and loss of assets, and contamination of the bay with belongings and chemicals washed out by the retreat of the water. Fortunately, due to the quick thinking of residents there were no fatalities and minimal injuries. (Vučetić et al, 2009). The LMTI allocated a Level 3.8 to this event (Level 3 bordering on Level 4) which is described in Table 2 as 'violent movement and damage to infrastructure and assets. Pollution by contaminants. Fast flowing velocity with deep water exceeding past flood risk defences'. The LMTI result and description accurately reflects the data for this event.

To demonstrate validation at the lower end of the index and a different geographical location the event at Arraial do Cabo (Brazil) on 7 September 2002 is given as an example. At 12:00 (UTC) and low tide, a series of unusual sea level oscillations at the maximum height of 0.7m occurred in the harbour. They were initiated by a sharp air pressure change (5 mb/hr) associated with an offshore weather system. Even though water velocity was strong, no damage to assets or injury/loss of life was reported (Candella and Araujo, 2021). The LMTI allocated this event a Level 2.2 which as described in Table 2 is an event that is 'visible in instruments and observations, causing slight disruption, with small debris and rarely a threat to life and assets.' The intensity level and description reflect the observed data for this event. However, even though the LMTIs ability to assess meteotsunami intensity was validated and demonstrated through this trial run, as the sample size is so small this will require further testing to ensure complete confidence.

Level 5 events are expected to be of a rare occurrence in the current climate. If one were to occur it would be distinguishable from the other levels by the extensive and long-term destruction of assets and loss of life that it would enforce on the shoreline.

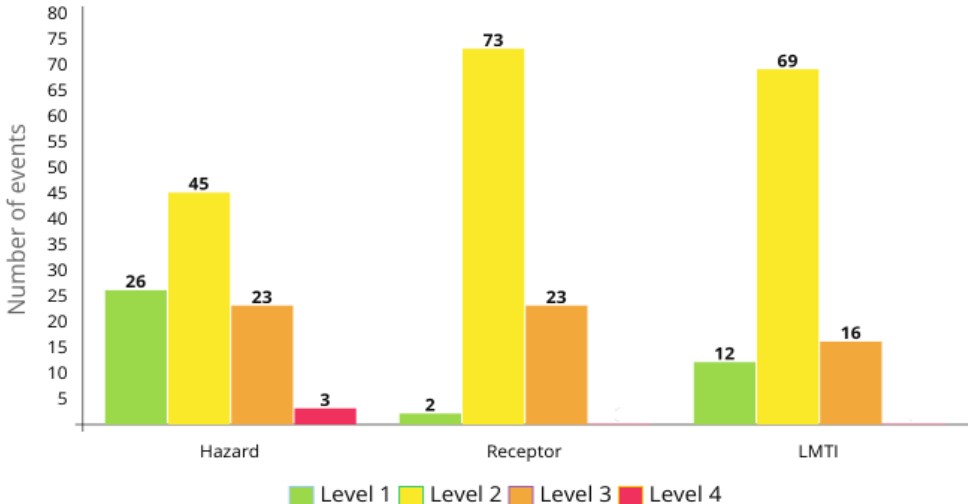

Figure 2: Hazard, receptor and LMTI index scores for UK meteotsunami (1750 to 2022).


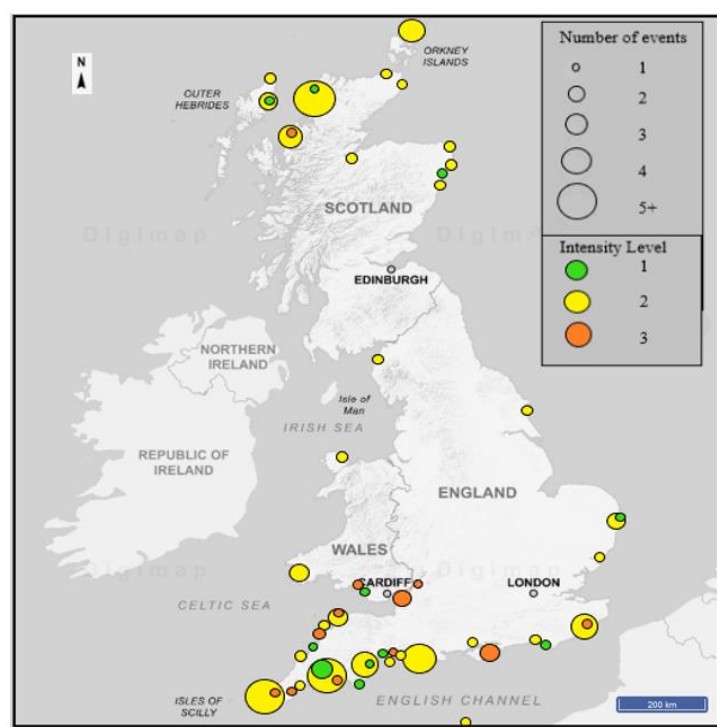

Figure 3: Geographical distribution of UK meteotsunami, with the number of events and the final LMTI intensity level shown at each location. Full results are in supplementary S1. Base map © Crown copyright and database rights 2022 Ordnance Survey (100025252).




**4 Discussion**



**4.1 The LMTI and UK meteotsunami**
Upon successful implementation of the LMTI in the UK, results have shown that meteotsunami have tended to be of moderate
intensity with an overall Level 2. Table 2 describes a Level 2 type event as representing visibly on instruments but rarely a
threat to life. Coastal communities will experience a slight disruption including flooding, the movement of small sized debris
and shallow water flow which will usually be accompanied by other hazards such as precipitation and lightning. The
identification of southwest England and Scotland as hotspots underscores the importance of the ability to run comparisons
between regions and events, allowing researchers to track changes in meteotsunami frequency, intensity and spatial distribution
over time. This hotspot tendency is most likely due to the dominant weather direction coming in from the west, off the Atlantic
Ocean and from strong convective storms building over Spain and France during the summertime.
The rareness of the combination of atmospheric, marine and topographical factors required for meteotsunami propagation is
why Level 4 (severe) events are small in quantity and observed at a limited number of locations. The strongest intensity
meteotsunami tend to appear in funnel shaped bays and harbours with a wide shelf which is necessary for Proudman resonance
to occur and the transfer energy from the atmosphere to the water. The western English Channel is sufficiently wide and deep,
with a shoaling coastline for meteotsunami to become well developed. The noticeable run of Level 3 and Level 4 hazard events
that occurred between 1883 and 1932 also coincided with a series of severe storms. The run of Level 1 hazard events between
2009 and 2015 are again due to a series of severe storms but in this instance, we can extrapolate a more accurate picture due
to the emergence of more refined quantitative data.
It is likely that the data for higher intensity meteotsunami events such as a Level 3 would have a more extensive historical
record compared to lower-level events such as a Level 1. This pattern can be attributed to the fact that major events tend to
have a more significant impact and are therefore more likely to be documented. The index has become more 'complete' over
recent years due to advancements in measurement and monitoring and an increase in the level of scientific interest and
awareness.

**4.2 Application of the LMTI index**
Motivated by the absence of a formalised way of quantifying meteotsunami intensity, in this paper we have presented the new
LMTI index which will allow for comparative analysis between regions prone to meteotsunami and it is offering a standardised
communication media to eliminate any confusion and inconsistency. Having started from a zero position the authors have
based the LMTI on the already widely accepted and used Papadopoulos and Imamura tsunami index (2001) and the ITIS-2012
tsunami index (Lekkas et al, 2013). Both indices are heavily reliant on qualitative perceptions based around the impact on
people and places. The latter, however, does incorporate quantitative data on the physical characteristics in the form of wave
height, run up and the number of fatalities, neither index accounts for variables such as resonance nor local geomorphology.
The traditional concept of an intensity scale measures the effects of hazards but not its strength (Gusiakov, 2009). The LMTI
has evolved this concept to incorporate the physical components of the hazard, this allows for further investigation into not
just why certain areas are more prone but also into the dynamics of the meteotsunami at the shoreline. The index is different
from other hazard indices as it does not require sophisticated technology and it allows for the analysis of both the hazard and
the receptor site to provide a more holistic view of meteotsunami risk. Understanding how these events have behaved and
evolved historically can be a precursor to establishing future trends and highlight issues to promote forward thinking in terms
of coastal planning. One of the primary strengths of the LMTI lies in its adaptability and potential for global application. As
the field of meteotsunami forecasting and warning progresses, the LMTI will no doubt play an important role in assisting in
this process. While the index was developed and trialled in the UK due to its long history of events, records and data
availability, it's underlying principles and methodology can be applied to other meteotsunami prone regions worldwide.

## 4.3 Constraints and limitations

While the expected results from the LMTI implementation are encouraging, there are certain limitations that should be
considered. The availability and quality of historical data may vary across regions, with events missing and the severity of
other events being underrepresented due to incomplete datasets, this may potentially affect the applicability in certain areas.
Addressing this limitation requires efforts to enhance data collection and establish robust monitoring networks.
The index contains two thresholds that rely on qualitative descriptors and many of the historical accounts used may have been
subjective in nature, especially with documents such as pamphlets and newspapers tending to misreport, exaggerate or invent
characteristics to boost sales. Results have revealed that the further back in time you go the less available and reliable the
accounts become. However, as time progresses this will be remedied with improved quantitative data collection methods.
Finally, sea level, shoreline slope and elevation in historical times would have been different from present day and the
geometric and topographic nuances of an area can have effects on the propagation of waves. As adjustment of this is beyond
the scope of this study; we must assume a static shoreline position based up on current data. Despite the limitations, the index
proves to be a useful indication of meteotsunami intensity, and these limitations should not be an issue in moving forward as
data becomes more available and at a higher frequency.

## 4.4 Further work

Successful implementation of the LMTI in the UK has yielded results that can be used to champion the need for higher
frequency data sampling on tide gauges and for the consideration of the inclusion of meteotsunami into coastal management
regimes. As this paper introduces the first evolution of the LMTI, we can offer potential strategies for calibrating and improving
the index, in particular for use in a more global context. Primarily, the incorporation of more data from recent observed events
and more global events will improve the calibration and reliability. The present evolution of the LMTI requires instantaneous
air pressure readings to indicate sudden changes. However, as wave height is proportional to integrated air pressure over time
it may be more appropriate to alter this component to incorporate air pressure over time.

Expanding the index to include resonances such as Proudman, Greenspan and harbour Q factor may provide valuable information about the potential amplification or dampening effects within a particular location. However, whilst this offers a valuable insight, it's practical implementation would require advanced numerical modelling techniques with reliable and detailed data on bathymetry, morphology and atmospheric conditions. This data may not be available for all affected locations and by adding complex resonances this could potentially hinder the practicality and usability of the index, making it harder to interpret and less accessible to decision makers.

## 5 Conclusions

After a review of the field of research for meteotsunami it was revealed that there was an absence of a standardised format for quantifying this phenomenon. In this paper, we have introduced a novel meteotsunami intensity index (LMTI), the first of its kind that mixes both quantitative data on the hazard with the effects on the shoreline. The successful implementation of the LMTI in the UK signifies an advance in meteotsunami research with results revealing a 69 % prominence of Level 2 (moderate intensity with slight disruption and a rare threat to life) type events occurring and the presence of distinct geographical hotspots in southwest England and Scotland.

Additionally, we successfully assessed the applicability and adaptability of the LMTI in a global context. As further trials and refinements are carried out, the LMTI has the potential to become a widely accepted standard, contributing to coastal planning and early warning systems worldwide.

**Supplement.** The supplementary UK map related to this article is available online at:
https://www.google.com/maps/d/edit?mid=1RiSeW-DIPSylIVOLv_8-T8Gy_e0To08&usp=sharing

**Author contributions.** C. Lewis developed the concept, designed, and executed the study and prepared the original draft. T. Smyth, J. Neumann, and H. Cloke supervised the project, provided advice, reviewed, and edited the manuscript.

**Competing interests.** The authors declare that they have no conflict of interest.

**Data availability.** The datasets used in this study were derived from resources available in the public domain.

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
