# Peer review of "Proposal for a new meteotsunami intensity index."

_EGUsphere, 2023_

## Author Response (AR1)

**Revisions: Comments and Authors responses.**

**R1** = reviewer 1, **R2** = reviewer 2, **C** = community, **A** = author

**R 1.1**: The manuscript attempts to quantify meteotsunami hazard and risk through a meteotsunami intensity index, LMTI, which is build based on common properties known for meteotsunamis. This is indeed a nice idea and approach, yet not easy to implement properly, so - in my opinion - more work should be done on that, i.e., on calibration of the index on the real meteotsunami events.

**A:** We agree with your suggestion that further work is needed to calibrate the LMTI index using a wider range of global meteotsunami events. We acknowledge that the implementation of the index is challenging, and we are committed to enhancing its accuracy by incorporating more data from observed events. In our revised manuscript, we will emphasise the need for continued calibration efforts and discuss potential strategies for improving the index's reliability.

**R 1.2:** Probably this index is much more adopted to the UK meteotsunamis, yet my problem is its global application, at least from what I see in the manuscript. Namely, L3 level of the index is given for all three events (Vela Luka in 1978, Ciutadella in 2006, Persian Gulf in 2017), to which I cannot agree knowing that there were 5 deaths and 17 injuries during the latter event, with measured waves (with 5 min sampling) higher than 2.5 m, with photos of the impressive tsunami wave advancing towards the shoreline, deeply inundated region, etc. (There are three research papers describing this event.) This event was among top10 deadliest and most hazardous known meteotsunami events, so I would expect to place it as L5 level, or at least L4 level, so I don't understand how it is qualified as L3 event.

**A:** We appreciate your concerns regarding the global application of the LMTI index, especially in light of the events in Vela Luka (1978), Ciutadella (2006), and the Persian Gulf (2017). If I may point you to the supplementary material, as you can see the LMTI has certain data gaps. After reevaluating these cases and submitting some of the missing data, the LMTI has been re calculated. The event of particular concern in the Persian Gulf (2017) was particularly deadly as it occurred in an area that was not accustomed to experiencing extreme wave events so consequently the infrastructure and the people were not prepared. Also, it occurred at 08.00 local time, a few hours after a thunderstorm and it was calm, so people were starting their day unawares of any problem. So, with the new data added the

event has now been upgraded to a Level 4 (4.1 to be exact) which is as you would have expected. This does, however, highlight the need for placing emphasis on a complete dataset to allow for a more robust scoring of intensity using the LMTI. We will revise the manuscript accordingly.

**R 1.3**: That might be due to relatively low tides in the region, that is also characteristics of the Mediterranean (and Vela Luka and Ciutadella cases). So, for such an index I would also took tidal range as the relevant variable, as the coastal infrastructure is basically adopted to it (e.g., in the Mediterranean) so, once you have the meteotsunami wave exceeding 10 times the tidal range, that should be different than if this ratio is 1 at maximum, like is probably for the UK sites where the tides are much larger.

**A:** We thank you for raising the issue of tidal range and its relevance to the LMTI index. While it is true that tidal range is an important factor in coastal dynamics and adaptation and that coastal infrastructure in certain regions, such as the Mediterranean, is adapted to the local tidal range, the effects of meteotsunami are modulated by such infrastructure, and the local topography. In micro tidal range areas while the wave energy lower it is more concentrated and in macro tidal areas even though the energy is a lot stronger there is a larger area exposed to dissipate the impact. Having carried out a provisional assessment of the tidal ranges in locations prone to different strength meteotsunami, we could not find a direct correlation. However, this does not mean there isn't one it just means that more analysis is required to understand and verify this variable. We have already incorporated a basic universal tidal component into the LMTI that at present is suitable to represent this issue. We will address this concern in the revised manuscript.

**R 1.4:** Another important issue missed by the authors is that there is an initial attempt to quantify intensity of meteotsunamis, based on their impact to the coastline, following the Papadopoulos-Imamura intensity scale developed for tsunamis. This is quoted in the referenced editorial by Vilibic et al. (2021), where they are suggesting meteotsunami intensity scale from 1 to 5, and meteotsunamis spatial coverage scale from 1 to 4. This scale should be at least substantially commented and discussed in this manuscript. For the Persian Gulf event in 2017, this scale is giving maximum intensity 5, following the deaths coming out of the meteotsunami.

**A:** We appreciate your suggestion to discuss the initial attempt at an intensity scale for meteotsunamis, as outlined in the referenced editorial by Vilibic et al. (2021). We will of course incorporate a substantial comment and discussion on this scale in our revised

manuscript. However, this scale as acknowledged by the author was limited to the events and papers represented in the special edition as a parameter that might be used for cataloguing meteotsunami. As we understand from the paper you have quoted there is no detailed methodology available, but the intensity scoring appears to be based upon wave height and injuries/fatalities. Measuring the intensity of meteotsunami solely based on fatalities is not scientifically robust because it does not provide a comprehensive understanding of the actual wave conditions. While fatalities can indicate the severity of an event, they are influenced by a range of factors beyond the wave intensity itself. Using fatality as a sole aspect would mean that a meteotsunami arriving on the shores of a highly populated area would indeed have more of an impact than an event occurring in a less populated area. It assumes that an event is only of high intensity if it has an anthropogenic impact. In some cases, even relatively small waves can result in significant loss of life if the affected population is caught unprepared or lacks proper warning systems.

Differentiating between hazard-based indices and impact-based indices is crucial for comprehensive risk assessment. While a standardised hazard index provides a starting point, it allows us to prioritise areas that are prone to specific hazards and allocate resources effectively to minimise the impact. It also allows for consistency and comparability across regions, facilitating resource allocation, policy development, and international collaborations. While hazard-based indices are useful, impact-based indices complement them by providing actionable information on the potential consequences of a hazard event, beyond human impacts alone. By incorporating both aspects the LMTI allows for more informed decisions and to take proactive measures to mitigate hazards' effects on both human and natural systems.

**R 1.5**: Further, there might be meteotsunamis not associated with air pressure jumps, but probably driven by strong wind fronts - for example, that is the case with so-called "winter-type meteotsunamis" in the northern Baltic Sea (Pellikka et al., 2022). So, it is a question if the air pressure only can be the only atmospheric variable which should be conjoined with definition of the LMTI index.

**A:** Thank you for your insightful comment regarding the consideration of only air pressure in the meteotsunami intensity index. We appreciate your perspective and acknowledge the existence of meteotsunamis that may be driven by strong wind fronts, as exemplified by the "winter-type meteotsunamis" in the northern Baltic Sea (Pellikka et al., 2022). While it is true that certain meteotsunamis can be influenced by factors other than air pressure, it is important to note that the focus on air pressure in the meteotsunami intensity index is justified by the

dominant role it plays in the generation and propagation of meteotsunamis in many regions worldwide. As stated in Pellikka et al, (2022) and Rabinovich (2020) air pressure jumps are found in other parts of the world during the strongest meteotsunamis. We agree that winter storms associated with meteotsunami can be conjoined with a pronounced mid tropospheric jet, however, this can manifest as an air pressure jump at the surface and low sea level pressure which can be seen in events in the Mediterranean, Japan, Australia and the Persian Gulf (Pellikka et al, 2022). Wind dynamics, such as intense gusts or abrupt changes in wind direction, can generate significant wave energy and as such in the current evolution of the index are considered as a compound hazard. While air pressure fluctuations have traditionally been associated with the generation of meteotsunamis, we concur that it is crucial to consider other atmospheric variables in future developments. As such in the revised manuscript, we will highlight the importance of investigating the role of additional atmospheric factors in the context of meteotsunami characterisation.

**R 1.6:** In conclusion, in my opinion it is quite hard to define an objective measure (like an index) of a meteotsunami, as they might come through different processes. Still, there are common variables connected with meteotsunamis, which obviously should be used there. For the UK meteotsunamis, the proposed methodology might work better, but for the world ocean more attempts should be done to properly calibrate to the observed events, in particular to these which were occurring in recent decades, and which have much more reliability and data for their description. Then it might be extended to the historical events, like for the UK. I would expect some comments in the discussion section along these lines.

**A:** We appreciate your recognition of the challenges in defining an objective measure, such as an index, for meteotsunamis due to their diverse generation processes. As the manuscript displays, the UK is a good starting point to trial the index due to its long history of events, records and data availability. However, it is important to have some form of a more global standardisation in quantifying meteotsunami. This will allow for comparative analysis between regions at risk and help create an effective communication media ruling out any confusion and inconsistencies. As you can no doubt appreciate this is a first attempt at such an index and the next steps will be to run more events to help calibrate to a more global context. Moreover, we agree with your suggestion that it is more prudent to focus initially on recent events with reliable data before extending the analysis to historical events.

Once again, we sincerely appreciate your valuable feedback and will expand the manuscript to include the points you have raised.

**R 2.1:** Firstly, it's not clear in the study what exactly counts as a meteotsunami. It is unclear whether authors examined the origin of each meteotsunami. Tsunami-like waves can be induced not only by atmospheric pressure, but also by infra gravity (IG) waves from meteorological phenomena such as windstorms, and for these cases, atmospheric pressure jumps may not be directly involved. An expanded discussion on the range of meteotsunami genesis would provide a more comprehensive foundation for the proposed index.

**A:** For the purpose of this manuscript and index we classify meteotsunami as shallow water waves which are initiated by sudden air pressure changes and wind stress from moving atmospheric systems such as convective clouds, cyclones, squalls, thunderstorms, gravity waves and strong mid tropospheric winds as defined by Vilibic and Sepic (2017). The distinction between various types of ocean waves and their sources can be complex and sometimes interconnected, so as far as we could proceed with the data available, we examined the origin of each meteotsunami before running it through the index. While we agree and acknowledge that infra gravity waves linked to strong mid tropospheric jets are correlated with meteotsunami genesis, this is restricted to such locations as the Mediterranean, Chile and Australia but not so much in the tropics (Zemunik, 2022). According to Demaniel et al (2023) infra gravity waves manifest as rapid surface pressure oscillations and low sea level pressure. This reinforces Rabinovich (2020) and Pellikka et al (2022) who state that air pressure plays the dominant role during the propagation of some of the world's strongest meteotsunamis as seen in the Mediterranean. Following on from this we chose air pressure as the dominant atmospheric component to cover both the mid latitudes and equatorial regions and to allowing for global standardisation of the index.

**R 2.2:** The proposed index was built on the integration of physician hazard characteristics and receptor site features. In my opinion, the proposed index can be useful to compare meteotsunami events at the same or nearby location, but the index in the current form is not appropriate to compare events from different locations. Addressing this limitation requires a further exploration of causality, correlation, and redundancy of parameters. It needs to be justified why the index uses the uniform weight for each parameter. It is imperative to discern the interplay and relative significance of each parameter and its contribution to the overall index score.

**A:** If I may refer you to Table 1 in the main document, where each parameter has a different threshold weighting leading to the allocation of a score of 1 to 5. These threshold weightings were calculated based on event data and other related hazard indices. As represented in

supplementary 2, the parameters have been tested on a range of global events where it has demonstrated its potential for use in other areas. We have already acknowledged to reviewer 1 that we agree the index will require further calibration by incorporating more data from global events.

**R 2.3:** The index lacks a parameter accounting for resonances, such as Proudman, Greenspan, and harbour/bay effects. The inclusion of such a parameter is pivotal for accurately quantifying meteotsunami intensity, as these resonances can significantly amplify wave impact.

**A:** While it's theoretically possible to include Proudman, Greenspan, and Harbour resonances in a meteotsunami intensity index, there might be some challenges and considerations:

Complexity: Including these resonances in an intensity index would likely increase its complexity. Meteotsunami generation and propagation are influenced by a wide range of factors, including local bathymetry, coastal geometry, atmospheric conditions, and more for which we have included components to reflect this such as shoreline elevation and gradient. Incorporating resonance effects may make the index harder to calculate and interpret.

Data Availability: As suggested in a previous reply to a community comment. Although resonances would contribute to a more accurate representation of meteotsunami on the local scale, accurate prediction and modelling of resonances would require detailed data on bathymetry, coastal morphology, and atmospheric conditions, which might not always be readily available for all regions where meteotsunamis can occur.

Practicality: The main goal of an intensity index is to provide a quick and effective way to assess the potential impact of a meteotsunami event. Adding complex resonances could potentially hinder the practicality and usability of such an index, making it less accessible for emergency response and decision-making.

**R 2.4:** The parameter "Fd" (Max Inundation Flooding) appears closely tied to local topography, potentially leading to correlations with "Sg" and "Se." The authors should investigate and clarify these potential correlations and determine if "Fd" remains suitable for comparing meteotsunami intensities across different locations.

**A:** Maximum inundation flooding is one of the factors that can contribute to the impact of a meteotsunami event. Including maximum inundation flooding in the intensity index allows for meaningful comparisons of meteotsunami events and provides a comprehensive and quantifiable measure of the potential damage and impact of a meteotsunami event. Researchers and emergency managers can use the index to rank events by their

potential impact based on the observed or predicted levels of flooding. Understanding the relationship between inundation flooding and meteotsunami intensity helps in assessing the overall risk posed by such events to coastal communities. This information is valuable for scientists, emergency responders, land use planners and policymakers working to mitigate the risks associated with meteotsunamis.

**R 2.5:** The importance of the parameter "Ch" (Number of Cumulative Hazards) may be heavily dependent on the locations. The severity of various phenomena like precipitation, mudflows, and lightning may be important at some locations (probably in the UK), but at meteotsunami "hot spots", this factor may be meaningless.

**A:** The number of cumulative hazards in a meteotsunami intensity index could be used to provide a comprehensive measure of the potential dangers posed by a meteotsunami event. Meteotsunami are not a stand-alone hazard, the source system brings with them other issues that contribute to the overall risk. I.e., precipitation will increase overall water levels and high winds will influence wave heights and direction. You cannot assess the impact without considering all the contributing factors. If this component is meaningless in some locations, then a zero can be placed in the calculation and this will not affect the overall intensity score.

**R 2.6:** The parameter "Ap" (Air Pressure) holds significance contingent upon the definition of a meteotsunami. Inclusion of this parameter will depend on the precise definition of a meteotsunami.

**A:** If I may refer you to AC1 above.

**R 2.7:** Parameters "Sm," "Sg," and "Se" exhibit potential importance, yet their relative weights remain unclear. Considering the possibility of redundancy among these parameters, assigning distinct weights to each would be prudent to avoid skewing results.

**A:** 'Sm' weightings are based on erosion capability of water, relative resistance, and the ability of the material to diffuse wave power, water currents and alter flow characteristics. The five weightings of shoreline material range from the least resistant material of a sandy beach (5 points); bedrock and gravel shores (4); estuarine and vegetated zones (3); artificial frontage such as concrete seawalls (2) and finally hard igneous rocks (1) which are dissipate wave energy and are a natural for of defence (Masselink et al. 2020 and Gornitz, 1991).

'Sg' and 'Se' weightings are linked to the susceptibility of the area to inundation and flooding by meteotsunami and can subsequently inform decision makers on mitigation factors

to be implemented. The thresholds for both components are adopted from the vulnerability index of Gornitz (1991) which is an already accepted and used methodology for assessing coastal hazards and risk.

If required it is possible to check for skew, correlation and redundancy amongst components (parameters) by performing various statistical tests. For skew analysis, well-established correlation coefficients such as Pearson and Spearman could be used and for guiding us to redundant parameters, variance inflation factor (VIF) could be implemented.

**R 2.8**: The presence of both "Ai" and "Fi" as damage parameters necessitates clarification, along with a more robust justification for the scale of "Fi." Moreover, the recently developed meteotsunami warning system may affect the parameter "Fi", and thus it may yield biased index values from different events at the same location.

**A:** These components were included to assess the level of asset impact as this is an intensity index which is a measure of the effects of a hazard not its strength (Gusiakov, 2009). These components were also included to allow for long term trend analysis, so if it is noted that fatality levels start to drop after the implementation of a warning system, then this would show that the warning system is effective especially if compared against an area with no such system in place.

**R 2.9:** L169: Allocating the same weight on the parameters does not seem to be reasonable.
**A:** Referring to Table 1, all 12 parameters (components) have distinct weightings. The equation implies that the 1 to 5 intensity score allocated to each component is divided by the number of components containing data (L165 – 168).

**R 2.10:** L247 – 257. This section needs to be written more precisely. For example, how can we say "independent" (L251)? The statements on L253-257 seem to be contradicting without any further explanation.
**A:** This can be addressed in the revision.

**R 2.11:** Figure1- It is unclear what this diagram implies. Furthermore, there are two arrows from "Document & data collection", and it is unclear which direction two follow.
**A**: This part of the diagram implies that there are two choices, either review previous meteotsunami events or review other hazard indices to find relevant components for inclusion in the index. This can be simplified if required.

**R 2.12:** The authors' proposed meteotsunami intensity index offers a promising framework for assessment. However, to establish its credibility and wide-ranging applicability, it is imperative to address the outlined concerns. By refining the meteotsunami definition, justifying parameter choices, accounting for resonances, investigating parameter causality, correlation and redundancy, and providing clearer guidelines, the proposed index could evolve into a valuable tool for evaluating meteotsunami events.

**A:** We acknowledge your concerns and will refine our definition and justify our parameter choices. In this paper we have introduced the first of its kind index for quantifying meteotsunami intensity. Developed from the already accepted and widely used Papadopoulous and Imamura tsunami index (2001) and the ITIS-2012 tsunami index (Lekkas et al, 2013), this index now allows for a similar formalised approach but with meteotsunami. Both tsunami indices are heavily reliant on qualitative perceptions based around the impact on people and place. The latter, however, does incorporate quantitative data on the physical characteristics such as wave height, run up and number of fatalities but does not account for variables such as resonance and local geomorphology.

The achievement of this index represents a significant advancement in aiding in the understanding and assessment of meteotsunami that until now, has experienced a lack of a standardized method for comparing this phenomenon. The LMTI fills this critical gap by providing a systematic and consistent way to quantify meteotsunami intensity.

**C.1:** It would be interesting to see how the index changes over time for the UK in a diagram to see the "completeness" of data over time. I imagine that LMT1 would only be "complete" in recent years, whereas LMT3 would be complete farther back in time. (Like how large earthquakes have a complete dataset farther back in time than small earthquakes).

**A:** The idea of visualising the changes in the meteotsunami index over time for the UK is indeed intriguing. This would provide insights into the availability and completeness of data over different periods, and it could potentially reveal interesting patterns or trends. In general, it is likely that the data for the higher level of meteotsunami events (LMTI 3) would have a more extensive historical record compared to lower-level events (LMTI 1). This pattern can be attributed to the fact that major meteotsunamis, like large earthquakes, tend to have a more significant impact and are therefore more likely to be documented in historical records. It's also important to recognise that the availability and completeness of data can also be influenced by various factors, such as advancements in measurement and monitoring techniques, changes in reporting practices, and the level of scientific interest and awareness over time.

**C.2:** It would be really interesting just to get a scope of LMT in different regions too.

**A:** Absolutely! Examining the scope of meteotsunami in different regions is the intended endeavour for future work.

**C.3:** How does the index round? Would you expect it to be written in whole numbers, or as LMT1 or LMT1.4 for example?

**A:** The presentation of the final meteotsunami index score is represented as a whole number, the index is typically rounded to the nearest integer. For instance, you might see the index represented as LMTI 1, LMTI 2, LMTI 3, and so on, without any decimal values. This simplified representation provides a clear categorization of intensity of meteotsunami events in which to present to the public, stakeholders and decision makers. In the calculation of the index, the scores are indeed expressed with decimal places as shown in supplementary 1. In this case, you encounter values such as LMTI 1.2, LMTI 1.5, LMTI 2.1, and so forth. The inclusion of decimal values enables a finer resolution in quantifying and comparing the intensity and impact of meteotsunami events for research purposes. In summary, the LMTI can be represented either as whole numbers or with decimal values. The choice of presentation format depends on the desired level of precision and the conventions followed by the researchers or organisations involved in studying and communicating meteotsunami events.

**C.4:** I'm not sure if I agree that a measurement of air pressure should be included. Air pressure magnitude can be completely different to the meteotsunami size, because wave height is proportional to the integrated air pressure over time (see equation 18 of https://link.springer.com/article/10.1007/s11069-020-03896-y), rather than any instantaneous air pressure. A pressure disturbance with linear growth/decay can produce the same size meteotsunami through Proudman resonance, with completely different values at the coastline (see Fig 9 of https://link.springer.com/article/10.1007/s11069-020-03896-y)

**A:** You raise a valid point regarding the inclusion of air pressure measurements in determining the intensity of meteotsunamis. The relationship between air pressure and meteotsunami size is indeed more complex than a simple instantaneous measurement. As it stands LMTI requires air pressure over time, this creates a connection with the mechanisms in the open sea, which can significantly affect the amplification or attenuation of meteotsunami waves. Also, by including air pressure data, researchers can identify

commonalities and differences between regions in relation to one of the forcing mechanisms. This comparative analysis may help refine the understanding of meteotsunami generation. In light of the research findings you referenced, it may be more appropriate to incorporate integrated air pressure measurements.

**C.5:** I really like how it mixes both quantitative data on the hazard (e.g. wave height) with effects (e.g. asset impact).
**A:** Thank you, we wanted to combine the two elements to enable an around view and to allow for the analysis of both the hazard and receptor.

**C.6:** Have you thought about including harbour Q-factor in the index?
**A:** Including the harbour Q-factor in the meteotsunami index is an interesting idea that could enhance the assessment of meteotsunami events. It could provide valuable information about the potential amplification or damping effects within a particular harbour. This would contribute to a more accurate representation of the meteotsunami on the local scale. While it offers valuable insights, its practical implementation would require advanced numerical modelling techniques with reliable and consistent data on the Q-factor values for various harbours, which may pose some challenges. It is however a consideration, moving forward.

.

---

## Referee Report (RR1)

Review on "Proposal for a new meteotsunami intensity index" by Clare Lewis, Tim Smyth, Jess Neumann and Hannah Cloke.

The development of the LMTI is a significant contribution to the field of meteotsunami research. This index provides a standardized and quantitative measure of meteotsunami intensity, addressing the need for a clear and consistent way to assess these events. The paper effectively highlights regional variations in meteotsunami occurrence and intensity within the United Kingdom. It points out the specific hotspots in southwest England and Scotland, which is valuable information for local authorities and disaster preparedness.

The main concern on this work is the validation of the index. An index is a systematic way to organize and access information or data. While the paper introduces an index and discusses its application to meteotsunami events in the UK, it does not describe a formal validation procedure. Validation is a critical step in assessing the reliability and accuracy of any index or model. It should involve comparing the results produced by the index with observed data to determine how well the index performs in different situations. Validation is particularly important when attempting to apply an index to various geographical regions or events because it helps confirm the index's robustness and applicability beyond its development context. In particular, it is unclear whether the index will be valid for the events of intensity 4 or 5.

Minor:
L270 Vilibic et als (2021) proposed intensity scale results -> intensity scale results proposed by Vilibic et al. (2021)
L328 under lying -> underlying

The graphical quality of Table 2 is poor. Increase dpi.

Supplementary
P3 Vela Luka Rtotal = 3.3 not 3.5 (But, this does not change the index score)
Some locations are missing the name of countries. For example, Ciutadella, Minorca
"Wido" is not in Japan.

---

## Author Response (AR2)

**Revisions: Comments and Authors responses.**

**R1** = reviewer 1, **R2** = reviewer 2, **E** = editor, **A** = author

**E:** One of the referees still has some concerns about the validation of the index. From my perspective, this is something that should be discussed a bit more in depth (so far, I only see a sentence in line 277) but that does not necessarily require additional analyses from your side.

**A:** Section added explaining using examples of how the index was validated against real world data.

**R2:** The main concern of this work is the validation of the index. An index is a systematic way to organize and access information or data. While the paper introduces an index and discusses its application to meteotsunami events in the UK, it does not describe a formal validation procedure. Validation is a critical step in assessing the reliability and accuracy of any index or model. It should involve comparing the results produced by the index with observed data to determine how well the index performs in different situations. Validation is particularly important when attempting to apply an index to various geographical regions or events because it helps confirm the index's robustness and applicability beyond its development context. In particular, it is unclear whether the index will be valid for the events of intensity 4 or 5.

**A:** Section added explaining using examples of how the index was validated against real world data.

**R1:** Just please correct Slavic letters (Šepić, Belušić, Mahović, Vilibić) and Demaniel (to Denamiel) in line 35.

**A:** Slavic letters added throughout document and name spelling changed.

**R2:** L270 Vilibic et als (2021) proposed intensity scale results -> intensity scale results proposed by Vilibic et al. (2021)

**A:** Sentence changed.

**R2**: L328 under lying -> underlying

**A:** Changed

**R2:** The graphical quality of Table 2 is poor. Increase dpi.

**A:** Graphical quality altered, and colours removed as per journals guidelines.

**R2:** Supplementary P3 Vela Luka Rtotal = 3.3 not 3.5 (But, this does not change the index score)

**A:** Calculation error changed.

**R2:** Some locations are missing the name of countries. For example, Ciutadella, Minorca "Wido" is not in Japan

**A:** Country names checked, all ok. Typo error for country of Wido changed to South Korea.